# Design and Synchronization Procedures of a D&F Co-Operative 5G Network Based on SDR Hardware Interface: Performance Analysis

**DOI:** 10.3390/s22030913

**Published:** 2022-01-25

**Authors:** Randy Verdecia-Peña, José I. Alonso

**Affiliations:** 1Information Processing and Telecommunications Center, Universidad Politécnica de Madrid, 28040 Madrid, Spain; ignacio@gmr.ssr.upm.es; 2ETSI de Telecomunicación, Universidad Politécnica de Madrid, 28040 Madrid, Spain

**Keywords:** decode-and-forward, co-operative 5G network, 5G-software defined radio, KPIs

## Abstract

Software defined radio (SDR) is a commonly used platform for its ease of operation and cost-effectiveness for the development and testing of real wireless communication systems. By supporting high transmission rates and enabling fast and cost-effective deployments, mainly in millimeter-wave (mmWave), the co-operative 5G network has been standardized by 3GPP Release 16. In this paper, a decode-and-forward (D&F) co-operative hardware network is proposed as one of the key technologies for future 5G/6G wireless networks. The proposed system consists of an emulated base station processing unit (gnodeB), a D&F protocol and the user equipment (UE). In particular, the design of the D&F relay node is based on an MIMO layer 2 relay technology. A testbed based on an SDR platform and Matlab${}^{\mathrm{TM}}$ software, in which the physical broadcast channel (PBCH) transmission, physical downlink control channel (PDCCH), physical downlink shared channel (PDSCH), and downlink shared channel (DL-SCH) for transport channel coding, according to the 3GPP standardized 5G downlink signal, has been designed. The key performance indicators (KPIs), namely EVM, BER, and throughput, were measured for 5G signals with 64-QAM and 256-QAM modulation schemes. The obtained results show that the D&F co-operative 5G network achieves substantially improved KPIs in the communication between the gnodeB and the UE in an outdoor-to-indoor scenario. Furthermore, it has been demonstrated that the D&F protocol presents a good performance and behavior being compared to one commercial equipment.

## 1. Introduction

The next generation of mobile networks (5G) is being deployed [1,2,3], which will bring new challenges and opportunities, enabling the creation and integration of new networks such as the Internet of Things (IoT), meeting the explosive growth in data traffic and lower latency requirements demanded in today’s communications and enabling and improving the quality of services in a multitude of applications, such as social networking, media streaming, video calls, and other broadband services [4,5,6,7,8,9]. In this sense, 5G cellular communication networks are expected to have a more reliable transmission link, better connectivity, and to approach the challenges involved with the mobility scenarios. The 3rd Generation Partnership Project (3GPP) has standardized the new radio waveform of the 5G new mobile generation [10,11], which proposes several ways for spectral efficiency and capacity improvement. The key enablers among the alternative approaches proposal can be found, for example, massive multi-input multi-output (mMIMO), ultra-dense network, millimeter-wave (mmWave), cell-free mMIMO, and co-operative networks, which can be achieved via the development of many access points of different types; that is why it is possible to have a higher resource block per unit area, improving the capacity of the user equipment (UE) and communication networks [12,13,14,15,16].

The co-operative architecture has been introduced and standardized within the 5G communication networks paradigm [17] by the 3GPP, for which the key benefits are enabling the flexible and very dense deployment of the new radio base station without increasing the cost-effective of its implementation. In this context, a diverse range of scenarios can be encompassed, as indoor-to-outdoor, outdoor-to-indoor, mobile urban, and rural environments [18,19,20,21]. It should be highlighted that the co-operative architecture for LTE communication network has been studied and standardized by the 3GPP [22], for which the most common relaying protocols proposed are the amplify-and-forward (A&F) and decode-and-forward (D&F) [23,24]. Nevertheless, due to the expensiveness of existing LTE mobile networks spectrum for backhaul link and small-cell deployments, it has been limited and restricted in its handful of commercial deployment in LTE timeline. co-operative 5G network is expected to achieve more commercial deployment than relay node co-operative LTE architecture due to mmWave communications, for which the key challenge is undertaken by the very high propagation loss being substantially limited by distance in comparison with LTE frequency band. co-operative architecture simple is composed of base station, relay node (RN), and UE, where the backhaul link is established between gnodeB and relaying technology, therefore, the access link is considered between relay node and UE, as been described in [16]. In-band and out-of-band backhauling with respect to the access link can be used to implement the co-operative network, and the relay node can support access and backhaul in sub-6 GHz (FR1) and above-6 GHz (FR2) spectrum, as has been defined by the 3GPP in [17]. Therefore, the main advantage to achieve a commercial success of the relay node co-operative network is the limited coverage of mmWave access, which creates a higher demand for backhauling deployment. In addition, the mmWave spectrum provides a more economically viable opportunity to deployment of wireless backhauling, being a key advantage over fiber [25].

At present, there are several testbench platforms to implement and evaluate the wireless communication technologies, which, with the Industry 4.0. is expected to increment demand to design and deploy the new nodes proposed by the 5G mobile communication network. Some platforms are based on field programmable gate arrays (FPGAs); however, their high cost hinders the usage by the research communities. The research communities have approached the software defined radio (SDR) platform to design and develop their wireless prototypes, which are more suitable for signal processing implementation through PHY and MAC layer functions [26,27,28]. Our investigation team has been designing and implementing wireless communication network through the software defined radio platform and Matlab${}^{\mathrm{TM}}$ software. Particularly, we have been deeply studying, designing, and implementing the relay node co-operative network over the downlink LTE signal [29]. Recently, refs. [30,31] designed and implemented a link-level out-of-band co-operative network simulator for the study of mmWave MIMO RN over downlink and uplink 5G signals, respectively. However, the real implementation of the co-operative network over 5G signal has not been deeply approached in literature yet.

### 1.1. Contribution

Motivated by the explained issues, in this paper, we consider the design and implementation of a co-operative 5G network with an in-band type 2 relay node for the downlink signal using a universal software radio peripheral (USRP) platform and Matlab${}^{\mathrm{TM}}$ tool. Therefore, the main contributions in this work, are summarized as follows:Design and implementation of a fully functional testbench, based on a MIMO software defined radio platform and Matlab${}^{\mathrm{TM}}$ software, which supports the following 5G downlink signal features [10,11]:Configuration and parametrization of the synchronization signal block (SSB), which supports the primary synchronization signal (PSS), secondary synchronization signal (SSS), and physical broadcast channel (PBCH) with its associated demodulation reference signal (DM-RS).Configuration of the variable signal bandwidth, CORESET, and search spacing functionalities.Physical downlink control channel (PDCCH) and its associated DM-RS.Physical downlink shared channel (PDSCH) and its associated DM-RS. In addition, enables the DL-SCH transport channel coding.These functionalities allow the generated 5G downlink signal from the emulated gnodeB to be demodulated by the MS2090A-Anritsu LTE/5G spectrum analyzer.Use of the developed architecture as an integrated platform supporting the radio access link in the 5G FR1 band, connecting a gnodeB and a relay node.An L2 relay node or decode-and-forward (D&F) relay node has been proposed and implemented. D&F node performs the functions of frequency synchronization, error correction, and search cell-ID stages, which allows practical adjustment of the received 5G downlink signal parameters. In addition, practical channel estimation by means of least square (LS) and minimum mean square error (MMSE) equalization techniques are introduced.A system model for the D&F node from the proposed hardware interface relay node co-operative network with real assumption is derived.Several measurement cases are carried out in real outdoor-to-indoor scenarios, using 64-QAM and 256-QAM modulation schemes for the 5G NR PDSCH.

### 1.2. Paper Outline

The paper is organized as follows. In Section 2, the technical background of system model for D&F relay node co-operative is introduced. Section 3 introduces the proposed D&F radio access hardware architecture and the completely functional testbench framework based on SDR and Matlab${}^{\mathrm{TM}}$ software. Section 4 describes the evaluated outdoor-to-indoor scenario. The obtained results and their explanation are carried out in Section 5, where the key performance indicators EVM (error vector magnitude), BER (bit error rate), and throughput are considered. Section 6 presents a summary of the obtained results and future works. Finally, Section 7 states the conclusions of this work.

### 1.3. Notation

The following notations are used in this paper: **x** ∈$C^{M\times 1}$ represents an M×1 complex vector, **X**∈$C^{M\times N}$ is an M×N complex matrix. $X^{H}$ is the Hermitian transpose of X matrix. $\left|\right|\cdot {\left|\right|}_{F}$ is the Forbenius norm. $N∼(m,\sigma^{2})$ describes a complex Gaussian random variable with mean *m* and variance $\sigma^{2}$.

## 2. Technical Background of D&F Co-Operative 5G System Model

The section presents the system model for 5G DL transmission via the proposed D&F co-operative architecture, as shown in Figure 1. The application scenario main of the proposed D&F access architecture is for sub-6 GHz indoor coverage. In this sense, D&F co-operative 5G network can communicate with both gnodeB and UE through backhaul and access links, respectively, as can be seen in Figure 1. The proposed D&F downlink co-operative 5G communication network comprises one gnodeB, an intermediate decode-and-forward relay node, and a destination (UE), which operates in a half-duplex (HD) mode. The gnodeB architecture is composed of two blocks mainly. Firstly, the baseband signal processing block is described by the $T_{\mathrm{BB}}$ matrix. This block considers a digital baseband recoder to configure the physical, control, and data channels, as well as the reference signals, taking into account the $N_{s}$ transmitted signals and modulation scheme. After that, the configured output signal from the digital baseband precoder is modulated by the 5G CP-OFDM modulator block, obtaining the downlink 5G waveform. The second block in the gnodeB considers RF transmit chains (FR1-Tx) to up-convert and amplify the baseband signal to a 5G frequency band, which in this work is 3.5 GHz. From the point of hardware, the interface has been developed through the SDR platform and power amplifier (PA), as will be explained later on.

In the proposed 5G co-operative network, D&F node introduces numerous advantages for effective 5G indoor coverage, where deployed as an intermediate node between outdoor source and indoor destination. D&F protocol comprises an array of receiver chains in sub-6 GHz (FR1-Rx) frequency band, for which the main function is to convert the FR1 band to baseband frequency band. The RF receiver stage ($M_{\mathrm{RR}}$) in the D&F is implemented through the receiver RF chains in the SDR platform. In addition, it comprises two baseband signal processing algorithms associated with the receiver ($R_{\mathrm{BBR}}$) and the transmit ($R_{\mathrm{BBT}}$) stages. Finally, the RF transmit (FR1-Tx) stage ($M_{\mathrm{RT}}$) in the D&F node up-converts the retransmitted baseband 5G signal to the FR1 (3.5 GHz) frequency band by means of the USRP platform. Then, in destination (UE) are inverted the operations performed by the source to estimate the received 5G signal through the D&F, which is achieved by the RF receiver chains (FR1-Rx) (it is implemented by means of USRP and described mathematically by $M_{\mathrm{DR}}$ matrix). Furthermore, the baseband matrix $G_{\mathrm{BB}}$ demodulates and decodes the 5G signal and NR channels, respectively.

Consider the 5G DL transmission over a co-operative wireless transmission channel, based on a D&F relay network system model, as shown in Figure 1, where a gnodeB has $N_{T}^{S}$, respectively, transmit antennas, power amplifiers, and RF chains. At the source, $N_{s}$ is number of data symbols (M-QAM modulation scheme), therefore, the modulated data symbols matrix is given by S∈$C^{N_{s}\times T}$, where T is the number of used subcarriers for the transmission (depending on bandwidth and subcarrier spacing). The $T_{\mathrm{BB}}$∈$C^{N_{T}^{S}\times N_{s}}$ baseband precoding matrix is used for modeling the control, data, physical, and reference signals as standardized by the 3GPP [10,11], as well as modulating the waveform of the 5G downlink. The gnodeB hosts $N_{T}^{S}$ RF transmitter (FR1-Tx) and is described by $M_{\mathrm{ST}}$∈$C^{N_{T}^{S}\times N_{T}^{S}}$ matrix. Therefore, considering the prior assumptions and a single-user (SU) downlink, the transmitted 5G downlink waveform from the source can be expressed as
(1)$$X=M_{ST}T_{\mathrm{BB}}S,$$
where power normalization is satisfied by $\left|\right|M_{\mathrm{ST}}T_{\mathrm{BB}}{\left|\right|}_{F}^{2}=TN_{T}^{S}$.

At the D&F node, $N_{R}^{R}$ and $N_{T}^{R}$, respectively, are receiver and transmitter antennas. Considering the proposed D&F node in Figure 1, the received signal processing is performed in two stages. First of all, the transmitted signal from gnodeB is received and decoded by the D&F protocol. After that, the detected data symbols are encoded and forwarded to the destination. It is worth highlighting that the proposed D&F is equivalent to a layer 2 LTE relaying technology, taking into account the 3GPP standardization and new release [17]. Therefore, the received signal matrix at the relay node can be written as
(2)$$Y_{\mathrm{DF}}^{1}=H^{1}X+N_{\mathrm{DF}}^{1}=H^{1}M_{ST}T_{\mathrm{BB}}S+N_{\mathrm{DF}}^{1}.$$

Here, $H^{1}$∈$C^{N_{R}^{R}\times N_{T}^{S}}$ and $N_{\mathrm{DF}}^{1}$∈$C^{N_{R}^{R}\times T}$ are the outdoor sub-6 GHz channel matrix between the FR1-Tx chains of the gnodeB and D&F node, and the additive white Gaussian noise (AWGN) matrix with zero mean and variance $\sigma_{N_{\mathrm{DF}}^{1}}^{2}$ at each antenna of D&F strategy, respectively. Furthermore, the $N_{R}^{R}\times N_{R}^{R}$ analog receiver $M_{\mathrm{RR}}$ matrix at the D&F node is composed of $N_{R}^{R}$ FR1-Rx chains. Hence, the signal $Y_{\mathrm{DF}}^{1}$ at the D&F protocol in Figure 1 after analog receiver matrix is
(3)$${\hat{Y}}_{\mathrm{DF}}^{1}=M_{\mathrm{RR}}^{H}H^{1}X+M_{\mathrm{RR}}^{H}N_{\mathrm{DF}}^{1}\;=M_{\mathrm{RR}}^{H}H^{1}M_{ST}T_{\mathrm{BB}}S+M_{\mathrm{RR}}^{H}N_{\mathrm{DF}}^{1}\;,={\hat{H}}^{1}X+{\hat{N}}_{\mathrm{DF}}^{1}$$
where ${\hat{Y}}_{\mathrm{DF}}^{1}$ = $M_{\mathrm{RR}}^{H}Y_{\mathrm{DF}}^{1}$. ${\hat{N}}_{\mathrm{DF}}^{1}$ = $M_{\mathrm{RR}}^{H}N_{\mathrm{DF}}^{1}$, and ${\hat{H}}^{1}$ = $M_{\mathrm{RR}}^{H}H^{1}$ are, respectively, the estimated noise and channel matrices between the gnodeB and D&F protocol.

As shown in Figure 1, the decoding and encoding stages in the D&F protocol are performed through $R_{\mathrm{BBR}}$∈$C^{N_{r}\times N_{R}^{R}}$ and $R_{\mathrm{BBT}}$∈$C^{N_{T}^{R}\times N_{r}}$ matrices, respectively. In this sense, once we have estimated the data symbols matrix ($\tilde{S}$) and consequently with D&F access architecture system, the transmitted signal by the D&F node can be defined as
(4)$$\hat{X}=M_{\mathrm{RT}}R_{\mathrm{BBT}}\tilde{S},$$
where $M_{\mathrm{RT}}$∈$C^{N_{T}^{R}\times N_{T}^{R}}$ denotes the IF→RF converter matrix in the relay node implemented by the RF transmitter chains of an USRP platform. In addition, $R_{\mathrm{BBT}}$∈$C^{N_{T}^{R}\times N_{r}}$ is the digital baseband precoder matrix in the transmitter side of the D&F node.

Let the received RF composite signal matrix at the user equipment be denoted as $Y_{\mathrm{DF}}^{2}$∈$C^{N_{R}^{D}\times T}$, which can be further expressed as
(5)$$Y_{\mathrm{DF}}^{2}=H^{2}\hat{X}+N_{\mathrm{DF}}^{2},$$
where $H^{2}$∈$C^{N_{R}^{D}\times N_{T}^{R}}$ is the frequency-wireless channel matrix between the D&F relay node and UE to transport $N_{r}\times T$ signals, and $N_{\mathrm{DF}}^{2}$$∼\mathrm{CN}(0,\sigma_{N_{\mathrm{DF}}^{2}}^{2}I)$ is the AWGN with noise variance in the access link $\sigma_{N_{\mathrm{DF}}^{2}}^{2}$.

UE is equipped with FR1-Rx chains and $N_{R}^{D}$ antennas. The number of FR1-Rx chains is the same as the number of receiver antennas, as shown in Figure 2. The RF→IF frequency converters at the UE perform downconversion of baseband frequency on the received RF signal $Y_{\mathrm{DF}}^{2}$. This stage is represented by the $M_{\mathrm{DR}}$∈$C^{N_{R}^{D}\times N_{R}^{D}}$. Therefore, at the UE, baseband matrix $G_{\mathrm{BB}}$∈$C^{N_{d}\times N_{R}^{D}}$ is employed to estimate the desired data symbols matrix being the inverse process to the performed in the base station and can be written as
(6)$$\hat{S}=G_{\mathrm{BB}}^{H}M_{\mathrm{DR}}^{H}\left(H^{2}\hat{X}+N_{\mathrm{DF}}^{2}\right)=\tilde{S}+N_{D},$$
where $N_{D}=G^{H}N_{\mathrm{DF}}^{2}$ and $G=G_{\mathrm{BB}}M_{\mathrm{DR}}$. From Figure 2, after the IQ is captured and downconversion of the baseband frequency is performed from the SDR platform hardware interface, the first stage in the $G_{\mathrm{BB}}$ is the frequency/time offset correction and SSB demodulation. For one-time obtaining the main parameters of the received 5G Signal as cell ID, the UE-node executes the full bandwidth demodulation. In this work, we have implemented the LS channel estimation and MMSE equalization as standardized by the 3GPP. Finally, it is performed decoding the control, physical, and data channels where $\hat{S}$ is obtained. In (Equation 6), the first term is the desired signal and the second term denotes the noise in access link at the decode-and-forward (D&F) relay node.

## 3. Testbench Implementation of D&F Co-Operative 5G Network

Our software defined radio platform and Matlab${}^{\mathrm{TM}}$ tool for co-operative LTE network was designed and explained in [29,32]. However, to implement the decode-and-forward co-operative network, the 5G system has been upgraded from the hardware interface, which will allow approaching research in the future of the out-of-band D&F node in the mmWave frequency band. The relay node co-operative wireless communications testbench is realized with three nodes, base station (gnodeB), relay node (D&F protocol), and destination (UE). The baseband processes are based on and implemented using Matlab${}^{\mathrm{TM}}$ software and realize the digital algorithms, for example, waveform generation and M-QAM modulation in the base station, channel estimation and equalization, decoding, and coding of the PBCH, PDCCH, PDSCH, and other channels in the D&F relay node. Furthermore, the demodulation and comparison of the received signals from base station and D&F protocol are performed in UE.

From the hardware interface, three separate universal software radio peripheral (USRP) devices, two NI-USRP 2944R, and one Ettus X310 are used as base station, D&F node, and UE, integrating digital/IF/RF units, respectively. The USRPs have tunable operating frequencies ranging from 70 MHz to 6 GHz and tunable transmission sample rates up to 200 mega samples per second. In order to assure a real-time relay node co-operative network implementation, we consider a specific computer configuration using three computers. Furthermore, USRPs can hold two RF chains (2Tx/2Rx). On the other hand, the connection from a PC to the NI-USRP 2944R/Ettus X310 is performed through the NI-IMAQdx GigE Vision High-Performance Driver, 10 gigabit ethernet card for desktop and 10 gigabit ethernet cable.

The upgraded SDR platform consists of one PA to the output of USRP in the base station, guaranteeing extending of the coverage of the co-operative wireless communication 5G network, as shown in Figure 3. In addition, the base station with PA achieves a distance of 49 m with 8% of EVM, which is the requirement by the 3GPP to the 64-QAM modulation scheme in a non-line-of-sight (NLOS) indoor-to-indoor scenario [33]. In addition, an OctoClock CDA-2990 has been added as an external source synchronizing multiple USRP devices (base station, D&F relay node, and UE) to improve the performance of the high channel count systems. On the other hand, the used antennas for the gnodeB node are R&S HL040E log-periodic broadband antennas. They work on 400 MHz to 6 GHz frequency band and their typical gain is 5.5 dBi. However, the D&F and UE nodes are equipped with Pasternack’s PE51RD1031 antennas, which cover the frequency band from 615 MHz to 4200 MHz and have a gain of 4.5 dBi.

Figure 4 shows transmitted signal demodulation, occupied bandwidth, channel power, and spectrum from the emulated gnodeB measured through the MS2090A-Anritsu spectrum analyzer to the output of the ZHL-1042J power amplifier. It can be seen that there is synchronization and modulation full between base station and Anritsu spectrum analyzer, which demonstrates the correct functionality of the emulated gnodeB (5G downlink signal). The synchronization signal consists of two parts, namely primary synchronization signal (PSS) and secondary synchronization signal (SSS), which in Figure 4 present 7.82% and 8.03% of the error vector magnitude (EVM), respectively. In addition, the physical cell ID (0) of the implemented gnodeB and its time and frequency structure of a single SSB have been detected, which includes the demodulation of the physical broadcast channel (PBCH), as shown in Figure 5. The transmitted frequency error is about 288 Hz, which is perfectly correctable with the implemented frequency offset correction algorithms in the D&F and UE nodes. The channel total power of the gnodeB is, approximately, 17.30 dBm. The obtained information from Figure 5 shows the root mean square EVM (%) of the PBCH, which can be seen to be considerably low, 2.60%.

### Proposed D&F Protocol Radio Access Hardware Architecture

Taking into account the general-purpose system model in Section 2 and hardware interface testbench described in Section 3, this subsection proposes a decode-and-forward relaying radio access hardware architecture. The proposed D&F protocol is flexible due to the implementation through Matlab${}^{\mathrm{TM}}$ software, as well as the encoding and decoding algorithms, which are adjusted to the 3GPP new radio standard [34]. In this sense, the 5G communication ToolBox of Matlab${}^{\mathrm{TM}}$ is employed. Furthermore, frequency division multiplexing (FDD) and MIMO antenna technique have been implemented, as well as in-band operation of the implemented protocol, that is, the backhaul and access links operate in the same frequency band, for which, in this work, the FR1 band (3.5 GHz) is considered.

The overall hardware interface of D&F relay node consists of two major steps, the radio frequency and baseband, as shown in Figure 6. In the first step, a 5G NR signal from the gnodeB is received and saved to file as complex in-phase and quadrature (I/Q). This step is performed by means of RF receive chains of the MIMO Ettus x310 SDR platform. On the other hand, in the second step, using the saved signal is performed in the baseband processing through the Matlab${}^{\mathrm{TM}}$ software. Considering the flowchart in Figure 6, the first block in baseband step is *NR Basic Structure*, which has input signal similar to the sample rate of SDR platform ($R_{p}^{c}$) and the received 5G signal (${\hat{Y}}_{\mathrm{DF}}^{1}$). The output of this block ($I_{\mathrm{OFDM}}^{\mathrm{NR}}$) supports the cyclic prefix, the desired number of iFFT points to use in the OFDM demodulator/modulator, the number of time-domain samples over which windowing and overlapping of OFDM symbols are applied, and the carrier frequency (Hz) allowing the phase precompensation applied for each OFDM symbol (as has been defined by the 3GPP in [11]). For initial access, the D&F protocol should start by the cell search procedure. By using the SS block or synchronization/PBCH block, being the most important task for synchronization of the nodes in the 5G network, the D&F relay node can estimate and correct the frequency and time offsets, as is shown in the CorrelationDetector block. It should be mentioned that the SSB consists of four consecutive OFDM symbols with 240 subcarriers each, where the primary synchronization signal and secondary synchronization signal are placed in the first and third OFDM symbols, respectively. In addition, the physical broadcast channel carries on the PBCH payload and PBCH-DMRS, as can be seen in the proposed flowchart, and is placed in the second, third, and fourth OFDM symbols, being modulated with the Q-PSK constellation scheme. In architecture, the PSS block detects the primary reference signal and determines the $N_{\mathrm{ID}}^{2}$. This detection has been performed employing the cross-correlation algorithm, in which the reference PSS sequence is created for the sector IDs (0, 1, and 2). Then, any significant frequency derivation must be estimated and eliminated, and the time offset must be performed without many errors being propagated. After the steps described previously, the 5G received signal is demodulated by means of the nrCP−OFDMdemodulation block.

Thereafter, the output of the PSS block is employed by the SSS block; thus, the detection of SSS in the received 5G signal and the estimation of the $N_{\mathrm{ID}}^{1}$ value is performed. Therefore, after both $N_{\mathrm{ID}}^{1}$ and $N_{\mathrm{ID}}^{2}$ are detected, and the physical cell ID ($N_{\mathrm{ID}}^{C}$) can be calculated as $N_{\mathrm{ID}}^{C}$ = $N_{\mathrm{ID}}^{2}$ + 3$N_{\mathrm{ID}}^{1}$ inside the CellIDDetection block. Now, with the known $N_{\mathrm{ID}}^{C}$ and RG (time/frequency resource grid), the PBCH DM-RS is extracted and correlated with each reference PBCH DM-RS. After that, the A channel and SNR estimations using PBCH DM-RS symbols for each possible DM-RS sequence are performed and estimated, respectively, in the same block. $\hat{H}$, $\hat{N}$, and RG are used to extract the received PBCH symbols from the SS/PBCH block, for which this step is implemented in the PBCHExtraction/MMSEEqualization block from the diagram, and MMSE equalization is performed on the PBCH symbols employing estimated channel and noise information after obtaining subcarriers of the PBCH. Then, the equalized PBCH symbols (${RG}_{\mathrm{RE}}^{\mathrm{PBCH}}$) are demodulated and descrambled in the NRPBCHDemodulation block. PBCH payload carries broadcast channel (BCH) payload which includes the master information block (MIB) and can be obtained through 24 bits of decoded PBCH transport block in the NRBCHDecode block. It should be noted that the D&F protocol obtains, by means of $I_{\mathrm{MIB}}$ (MIB), the information of the system frame number (SFN), the subcarrier spacing for system information block 1 (SIB1), and the frequency offset between SSB and resource grid.

Once the signal parameters are known, the received signal (${\hat{Y}}_{\mathrm{DF}}^{1}$) is resampling to the nominal sampling rate and is adjusted to the frame origin through timing offset, which is performed from $T_{c}^{o}$(·) function in the NR decode-and-encode Algorithm 1. This correction step has been achieved by performing timing estimation by cross-correlation with the known reference signals. Therefore, the CP-OFDM demodulation on full bandwidth is realized by the ${nrCP}_{d}^{\mathrm{ofdm}}\left(\cdot \right)$ function, where the $N_{\mathrm{ID}}^{C}$, $I_{\mathrm{MIB}}$, and $I_{\mathrm{OFDM}}^{\mathrm{NR}}$ are required as input parameters. In order to minimize the effects of noise and channel distortion, Algorithm 1 implements MMSE equalizer over the demodulated signal (Y), which has been developed considering single and multiple antennas. After the steps described above, Algorithm 1 is in condition to execute the decode and encode of the physical, control, and data channels, as well as determining other reference signals of the 5G downlink signal in each time slot ($N_{\mathrm{sl}}$). It should be highlighted that the low-density parity-check (LDPC) technique has been considered to decode and encode the DL-SCH channel, fulfilling the 3GPP standard [35]. Besides, in the proposed algorithm, the sub-indices ${\left(\cdot \right)}^{d}$ and ${\left(\cdot \right)}^{c}$ in the functions describe the processes of decoding and encoding, and the sub-indices ${\left(\cdot \right)}^{b}$ and ${\left(\cdot \right)}^{s}$ in the outputs of the functions represent data bits and data symbols, respectively. The ${NR-P}^{c}$ function, mapping each channel in the resource grid, and the $\hat{x}$ subframe to the output of the function, is obtained. Accordingly, concatenating each subframe is performed by mean of the Cat(·) function, from which all frames retransmitting are saved. $\hat{X}$, the resultant signal to the output of Algorithm 1, is modulated by the nrCP−OFDMModulate block. After that, an appropriate portion of SS burst is added to the modulated signal. Then, the resultant waveform is transmitted by means of RF chains of the SDR platform to UE, as shown in Figure 6.
**Algorithm 1:** NR Decode-&-Encode Algorithm

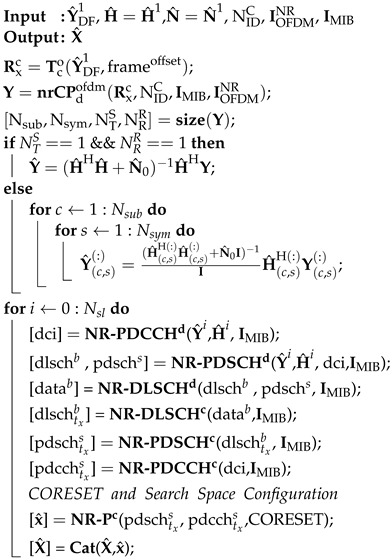


## 4. Experimental Setup

This section describes the practical scenario where the proposed D&F relay node co-operative 5G network has been executed and verified. Figure 7 shows the scenario that has been created with the goal of validating the proposed D&F protocol co-operative hardware interface in the context of a specific use case (outdoor-to-indoor).

A series of real field measurements have been performed in *E.T.S.I. de Telecomunicación, Universidad Politécnica de Madrid*, in particular in an outdoor-to-indoor environment, on the hall of the fourth floor of Building C and 406 office, as can be seen in Figure 7. The experimentation scenario is composed of one gnodeB, which will serve the one UE through the D&F relay node. In addition, the base station is physically available at the hall window of the front courtyard of Building C, as shown in subfigure of Figure 7. The backhaul link is represented in Figure 7 with a red arrow, which transmits the 5G downlink signal over antennas that are outside of the building. In the scenario, we consider that gnodeB and D&F protocol have line-of-sight (LOS) and the distance between them is approximately 55 m. Therefore, the data information is received by the relay node employing other antennas placed outside of the opposite building, specifically in the window of the 406 office. Therefore, the received 5G signal is processed in the D&F protocol and retransmitted to the UE inside the 406 office through the access link, which is represented by the light blue arrow in Figure 7. It should be mentioned that D&F relay node and UE have a LOS channel and a separation distance of 3 m. Finally, a third link has been established between gnodeB and UE nodes, named as direct link, which has non-line-of-sight (NLOS), and the distance between them is 58 m. It should be highlighted that the idea behind the third link is to compare the performance over a 5G downlink signal between the relay co-operative system (gnodeB-to-D&F-to-UE) and a direct communication (gnodeB-to-UE). Consequently, the UE calculates the KPIs metric for the two kinds of link-levels individually and performs the comparison between them.

## 5. Experimental Results

In this section, we evaluate the performance of the synchronization procedure in the D&F relay node over 5G downlink signal in terms of error vector magnitude. After that, the KPIs of the proposed decode-and-forward protocol access hardware architecture in terms of bit error rate (BER) and throughput versus $P_{BS}$ (transmission power of base station) for 64-QAM and 256-QAM modulation schemes have been taken and analyzed. Note that the KPIs are evaluated from the downlink obtained through a gnodeB emulated employing the Matlab${}^{\mathrm{TM}}$ 5G ToolBox, where for each transmission (in this work 20), 80 frames were sent by the base station for each point of $P_{BS}$. Furthermore, we consider a 5G downlink signal (CP-OFDM modulator) with subcarrier spacing 15 kHz, a sampling rate of 7.68 MHz, and a carrier frequency of 3.5 GHz. The rest of the emulated gnodeB parameters are given in Table 1.

### 5.1. Synchronization and Performance of the Backhaul Link in the D&F Relay Node

In this subsection, the performance of the developed synchronization stages in the D&F relay node are evaluated. The first task of the D&F protocol after capturing the received signal through the SDR on the backhaul link is cell search, which is achieved by decoding the SSB to establish the synchronization between gnodeB and D&F relay node. In this sense, D&F protocol finds out time and frequency synchronization with a cell and decodes the cell ID of the gnodeB. Thus, relay node decodes the PSS and SSS to determine the physical cell ID. Figure 8 shows the PPS correlation versus frequency offset (kHz) for three sector IDs 0, 1, and 2. From the figure, it can be seen that $N_{\mathrm{ID}}^{2}=\;$0 was selected due to that it has the highest magnitude. In the next step, decoding of the secondary synchronization signal is processed, for which SSS correlation is shown in Figure 9. Therefore, the cell ID group ($N_{\mathrm{ID}}^{1}$) with the highest correlation is determined, in this case, the cell ID group 0 was selected. Now, the D&F protocol can know the cell ID ($N_{\mathrm{ID}}^{C}$) by 3 × 0 + 0 = 0. In order to further verify the developed synchronization algorithms, the values obtained before are compared with those obtained with the Anritsu spectrum analyzer. As can be seen in Figure 4 and Figure 5 in Section 2, both values are 0. After that, the strength of signal for PBCH demodulation reference signal (DM-RS) for each set of SSB is detected (in this paper the authors have considered one SSB by frame) by the D&F relay node.

Figure 10 shows the PBCH DM-RS estimated SNR versus iSSB, where the index of PBCH DM-RS with the highest correlation (0) defines the LBSs of SS/PBCH block required for PBCH scrambling initialization. This beam is also identified as the best beam for the D&F protocol.

Once successful decoding information by the D&F protocol is achieved, downlink control and the physical channels are now enabled by the obtained data symbols of the PBCH, in which the required information ($I_{\mathrm{MIB}}$) is obtained to demodulate these channels. In this sense, Figure 11 shows the downlink constellation diagram for the physical broadcast channel. The left subfigure represents the received signal without equalization technique, and the right illustrates the equalized data symbols after applying the MMSE equalization algorithm. It should be noted from Figure 11 that the MMSE technique introduces significant improvement over the received PBCH symbols. Furthermore, it should be highlighted that the previous measurements have been performed considering the parameters of Signal 1 and the scenario of Figure 7. On the other hand, experimental results have been obtained, taking into account signal 2, and similar results have been drawn.

In this campaign, we also assess the control and data channels behavior of the received D&F relay node signal by the backhaul link. For each channel, the EVM in dB for the received symbols is measured by
(7)$${\mathrm{EVM}}_{\mathrm{ch}}\left[\mathrm{dB}\right]=20\times \log_{10}\left[\frac{\sum_{n}^{N_{s}}{\left|{\hat{S}}_{s,n}-S_{s,n}\right|}^{2}}{\sum_{n}^{N_{s}}{\left|S_{s,n}\right|}^{2}}/100\right],$$
where ${\hat{S}}_{s,n}$ is the decoded symbol s matrix at the D&F protocol and $S_{s,n}$ describes the originally transmitted data or control symbol s, s indexes the subcarrier position and n is the OFDM data symbols index, the median EVM of the PBCH obtained at D&F relay node by two signals (1 and 2 Table 1), LOS and NLOS channels, and equalization step is shown in Figure 12. In addition, the measured PBCH through the Anritsu spectrum analyzer is employed for comparison. It should be highlighted that the PBCH is transmitted by the Q-PSK modulation scheme. It can be seen from Figure 12 that lower transmit power levels at the base station translate into a lower desired signal quality at the D&F relay node. In this sense, lower signal-to-interference noise ratio (SINR) induces higher EVM for the data symbols of the evaluated channels. On the contrary, the SNR increases for higher power levels $P_{BS}$ at the gnodeB, therefore enhancing the EVM results at the PBCH of the D&F protocol, as is shown in Figure 12.

It should be explained that the considered NLOS backhaul link corresponds to the scenario in Figure 7, where a non-line-of-sight between base station and relay node has been established. The aim of this experimental environment is to verify the functioning of the synchronization and decoding steps in the D&F protocol under a scenario with adverse communication conditions. It can be observed, from Figure 12, that as expected, the NLOS experimental scenario presented worse EVM (dB) than the LOS environment. Furthermore, the developed MMSE equalization method in the relay node considerably improves the performance of the decoded received symbols at the PBCH. For example, when considered the signal 1, LOS scenario, and $P_{BS}=7$ dBm, the EVM of the PBCH after the MMSE equalization technique is approximately −38.29 dB; however, before the MMSE technique it is −35.98 dB; therefore, it can be appreciated that it introduces a significant improvement on the order of 2.31 dB. On the other hand, to verify the obtained results through D&F protocol, two new graphics have been inserted (black and blue) for its comparison, as is shown in Figure 12. The obtained information from the figure shows that the results with the Anritsu spectrum analyzer have similar behavior to the proposed algorithms of the implemented D&F in both scenarios, LOS and NLOS channels. The authors have also measured the physical downlink control channel in the D&F relaying technology, and the obtained results are drawn in Figure 13. The symbols of the downlink control channel are transmitted by the Q-PSK modulation scheme. In addition, from the figure, it can be seen that the performance behavior by the decoded PDCCH in the relay node is similar to that achieved in Figure 12, where the best performance is reached with the LOS scenario and post-MMSE technique.

Figure 14 shows the EVM results at physical downlink shared channel for 64-QAM and 256-QAM constellations, which correspond to signal 1 and signal 2, respectively. It can be seen that the EVM (dB) increased when the NLOS channel was considered. Moreover, the MMSE technique significantly improved the system performance, compared to the measured pre-MMSE algorithm. For example, when signal 1, LOS environment, and MMSE equalization were taken into account, the PDSCH of the D&F relay node achieved an EVM (dB) level of −42.19 dB at around $P_{BS}=15$ dBm while, without the MMSE technique, the PDSCH of the relay node reached −33.43 dB for the same $P_{BS}$, such that a performance gain of 8.76 dB was achieved by the MMSE technique. Furthermore, comparing the results of Figure 14, it can be observed that the 64-QAM signal constellation presented less error than 256-QAM in both measurement environments. In the case of the LOS channel and $P_{BS}=7$ dBm, a 1.85 dB performance gain was successfully achieved by the PDSCH 64-QAM when compared to the EVM of the PDSCH 256-QAM modulation scheme.

### 5.2. Performance Evaluation of Proposed Hardware D&F Co-Operative 5G Network

This subsection shows experimental results for the D&F relay node co-operative 5G network in terms of the bit error rate and throughput of the 5G NR physical downlink shared channel transmission from base station. In the results, the proposed D&F protocol is compared with non-co-operative 5G network (direct link). Furthermore, we consider an experimental network encompassing one gnodeB equipped with $N_{T}^{S}=\left\{1,2\right\}$ transmitter antennas, and transmitting the PDSCH over 64-QAM and 256-QAM modulation schemes. It should be highlighted that, in the figures, D&F relay node represents the received signal of the UE through the access link, and direct link symbolizes the waveform received at the UE directly from gnodeB. On the other hand, the D&F protocol is equipped with $N_{R}^{R}=N_{T}^{R}=\left\{1,2\right\}$ receiver and transmitter antennas; futhermore, the user equipment is equipped with $N_{R}^{D}=\left\{1,2\right\}$. We have denoted $N_{T}^{S}\times N_{R}^{R}=N_{T}^{R}\times N_{R}^{D}$ as per the considered MIMO antenna technique in the proposed D&F relay node co-operative 5G network.

The first results have been performed through the Anritsu spectrum analyzer, in which the main aim is to demonstrate that the retransmitted signal by the D&F relay node can be demodulated by the commercial equipment and to verify the functioning correct of this stage on the proposed node. In addition, direct link has been measured for comparison. The PBCH demodulation and 5G summary of the signal transmitted by the gnodeB are shown in Figure 15, in which direct link has been taken into account. Moreover, Figure 16 shows the same results but considering the received signal from the access link (D&F protocol). Therefore, from the obtained results in Figure 15 and Figure 16, it can be observed that there exists a full synchronization and demodulation in both links. Furthermore, it should be noted that the UE achieved better demodulation of the PBCH through the received signal from D&F relay node with 6.33% of RMS EVM. However, the PBCH demodulation is reached by means of direct link but with higher EVM, exactly 28.77%. Other channels and reference signals have been detected and their average EVM are shown in Figure 15 and Figure 16, for example, PSS, SSS, and SS-SINR of the direct link with 35.20%, 32.39%, and 9.79 dB, respectively. Nevertheless, the obtained results of the EVM for access link are considerably low in comparison with direct link, 9.40% and 9.47%, respectively. Moreover, the SINR reached though the D&F protocol was 20.47 dB with a gain of 10.68 dB with respect to a direct connection between gnodeB to UE.

The rest of the discussed results have been obtained through the implemented UE. Therefore, Figure 17 shows the corresponding BER performance comparison of the implemented D&F relay node co-operative 5G system and non-co-operative 5G network, taking into account the 64-QAM modulation scheme over the PDSCH. It can be seen that direct link with SISO achieved a discrete performance. However, the D&F protocol co-operative 5G network with SISO technique reached the best performance among the implemented system with SISO technology. From Figure 17, it can be verified that the direct link with SISO antenna technique reached the BER level of 1.5 × 10${}^{-1}$ at around $P_{BS}=15$ dBm. Nevertheless, when the MIMO antenna technology was introduced in the non-co-operative system, specifically 2 × 2 MIMO, it can be seen that the direct link increased the BER performance, in comparison with the SISO technique. In this regard, we note that the 2 × 2 MIMO non-co-operative network achieves the same BER level of the SISO direct link at around $P_{BS}=11$ dBm so that a performance gain of 4 dBm is obtained. Moreover, when SISO D&F relay node co-operative network is taken into account, a BER level of 2.7 × 10${}^{-2}$ is reached, which supposes an improvement of 4.2 dB in comparison with 2 × 2 MIMO non-co-operative network. Nevertheless, the (2 × 2 × 2) MIMO D&F protocol co-operative system has generated a significantly improved behavior, as can be seen in Figure 17. From the figure, it can be concluded that (2 × 2 × 2) MIMO D&F relay node acquired a BER performance of 1.2 × 10${}^{-3}$ at around $P_{BS}=15$ dBm, implying that 9 dB and 6.7 dB performance gain was successfully reached, in comparison with SISO and 2 × 2 MIMO non-co-operative network, respectively. In addition, (2 × 2 × 2) MIMO D&F network overcomes the performance of the SISO D&F protocol by approximately 3.5 dB.

Figure 18 illustrates the achievable BER performance of the proposed downlink relay co-operative 5G network, in which the 256-QAM modulation scheme has been considered. It is indicated from Figure 18 that the implemented downlink 2 × 2 MIMO non-co-operative network achieves a lower BER level than the same developed structure with the SISO antenna technique. More explicitly, the error of the 2 × 2 MIMO non-co-operative decreases, and the BER level of this structure improves and approaches the 1.5 × 10${}^{-1}$ BER level at around $P_{BS}=15$ dBm, which suggests one gain of the 3.8 dB of the downlink SISO non-co-operative network. Additionally, when the D&F protocol co-operative system has been taken into account, the BER level is decreased considerably, as can be seen in Figure 18. Therefore, for the downlink SISO D&F protocol, an 8 dB performance gain was attained and overcome, with $P_{BS}=8$ dBm in the SISO non-co-operative network. Furthermore, the results clearly show that the SISO D&F relay node has better performance than 2 × 2 MIMO non-co-operative structure, for example, the same behavior as the 2 × 2 MIMO non-co-operative has been obtained at $P_{BS}=9$ dBm, such that a 6 dB performance gain was achieved. As shown in Figure 18, the BER performance of the (2 × 2 × 2) MIMO D&F protocol decreases tremendously, which, compared with MIMO non-co-operative and SISO D&F relay node networks, obtains an enhancement of 8.1 dB and 4 dB approximately, respectively. It should be also highlighted that the BER performance of the 64-QAM is lower than 256-QAM, which was due to the robustness that changes the 64-QAM constellation scheme to 256-QAM.

In Figure 19 and Figure 20, we examine the achievable capacity performance of the proposed downlink D&F protocol co-operative 5G network with single and multiple antenna techniques for the 64-QAM and 256-QAM constellation schemes. Therefore, the illustrated performance in Figure 19 and Figure 20 describe the output throughput of the new radio DL-SCH transport channel processing measured by the user equipment, respectively. From the figures it is shown that the achievable capacity performance recorded for the downlink MIMO co-operative and non-co-operative networks reached a higher average throughput than SISO antenna technology in the same environments. It can be seen from Figure 19 that the maximum throughput obtained by the SISO non-co-operative network was 3.27 Mbps; however, with 2 × 2 MIMO, a capacity of 4.78 Mbps could be reached with the same $P_{BS}=15$ dBm, which represents a performance gain of about 1.51 Mbps. Nevertheless, when the SISO D&F protocol is considered, the average performance of the network is so much higher than the 2 × 2 MIMO non-co-operative system—for example, the reached maximum throughput was 8.95 Mbps, which achieved a gain of 5.68 Mbps and 4.17 Mbps with respect to SISO and MIMO non-co-operative networks, respectively. It may also be seen that the throughput of the (2 × 2 × 2) MIMO D&F relay node co-operative system at $P_{BS}=10.3$ dBm was approximately 8.95 Mbps, corresponding to the maximum value reached by SISO D&F protocol, such that a performance gain of about 4.7 dBm was achieved. Furthermore, the obtained results in Figure 19 have been observed as a substantially improved system throughput of the D&F relay node with the MIMO technique in comparison with direct link, in which performance gains of 7.80 Mbps and 6.29 Mbps were reached, respectively, with respect to SISO and MIMO techniques. Finally, it should be noted that the maximum throughput of our downlink 5G NR for the 64-QAM constellation scheme is 11.066 Mbps. Therefore, from Figure 19, it can be seen that this value was reached by the (2 × 2 × 2) MIMO D&F protocol co-operative system from $P_{BS}=$ 13 dBm.

Here, the experimental results of the hardware interface downlink D&F relay node co-operative 5G network are given for 256-QAM modulation scheme over SISO and MIMO channels, as shown in Figure 20. It can be observed that the downlink SISO direct link reached a capacity of 1.87 Mbps at around $P_{BS}$ = 15 dBm, while with the 2 × 2 MIMO antenna technique, the non-co-operative network reached the same average throughput at around $P_{BS}$ = 11 dBm; accordingly, a performance gain of 4 dBm was achieved. The maximum capacity of the 2 × 2 MIMO direct link was 6.12 Mbps, but it is very far off the throughput reached by the D&F relay node. In this sense, when the SISO D&F relay node co-operative 5G network was taken into consideration, the user equipment reached 10.47 Mbps at around $P_{BS}$ = 15 dBm, implying that a 3.6 dB performance gain and 4.35 Mbps were successfully achieved compared to the 2 × 2 MIMO direct link performance. It should be highlighted that the tremendous enhancement with the co-operative network is due to the impact of decoding and encoding stages the downlink FR1 received signal was subjected to before being retransmitted to the UE by means of the downlink D&F protocol. Additionally, it should be noted that the throughput maximum that could be reached without any channel for the 5G new radio PDSCH 256-QAM system (signal 2 of Table 1) is 15.206 Mbps, which only was achieved when employing the (2 × 2 × 2) MIMO D&F relay node at $P_{BS}$ = 15 dBm. In this context, it can be observed from the figure that D&F protocol with MIMO antenna technique achieved approximately the same average throughput as the SISO D&F relay node at 10.8 dBm. However, with respect to non-co-operative network, a gain of about 13.34 Mbps and 9.09 Mbps, respectively, for SISO and MIMO was obtained.

## 6. Discussion and Future Works

In this work, a co-operative 5G network was designed and implemented, based on the SDR hardware interface and Matlab${}^{\mathrm{TM}}$ software, using a D&F protocol as an intermediate node to facilitate communication between source and destination. A decode-and-forward protocol symbolizes a layer 2 relay technology, in which the RF signal from the base station is first decoded, and then forwarded to the destination node where it is again encoded. It is worth noting that the 5G signal emulated in the D&F co-operative SDR platform and the developed algorithms comply with the requirements standardized by 3GPP, being one of the main novelties of the presented work.

Looking at the results of the backhaul link measurements, it can be seen that the algorithms implemented in the D&F protocol have an excellent behavior for the synchronization, correction, decoding, and coding of the 5G signal received from the gnodeB. In this respect, it should be noted that the obtained results are closely related to those obtained with the MS2090A-Anritsu LTE/5G spectrum analyzer (see Figure 16). Besides, the analysis of the EVM over broadcast channel, control data channel, and downlink shared channel shows that the MMSE equalization stage decreases the error magnitude, which improves the system performance. It is worth noting that the EVM, before the equalization stage, does not decrease as the transmitted power increases, which is due to the fact that as the $P_{\mathrm{BS}}$ increases, the noise at the reception, in the experimentation scenario (Figure 7), is also amplified; however, when using the MMSE technique, the noise is considerably canceled out. In addition, the implemented algorithms were verified over controlled lab environment [33], obtaining one EVM of the PBCH which decreases as transmitted power increases.

The results obtained in previously published works [29,36,37,38,39] are consistent with those obtained in this paper, demonstrating the improvements that the use of D&F relay nodes introduces in the performance of the network. Table 2 shows a comparative summary of the experimental results obtained in this paper. The presented results show the advantages of inserting D&F relay node in 5G mobile networks. Furthermore, it should be noted that in the case of mmWave communications, the use of relay nodes will be more necessary, as demonstrated in [30,31].

Future research work includes the development and implementation of algorithms for the uplink of the 5G signal. This would provide a co-operative 5G network with D&F relay nodes working bidirectionally (uplink and downlink). In this context, we will address the implementation of new algorithms to develop the hardware interface of the SDR platform for 5G uplink communication and upgrade the 5G signal to the 20 MHz bandwidth to improve the capacity of the co-operative network, as the SDR platform supports the requirements. Another future objective, which is currently being worked on, is the development of transceivers in millimeter bands (26 GHz), which will allow the implementation of D&F relay node in these frequency bands. This will allow analyzing the feasibility, through theoretical studies and measurements, of the use of relay nodes in some of the vertical markets proposed by 3GPP, such as high-speed or high-vehicle-density transport. These works are a first step to analyze the advantages and disadvantages of using IAB (integrated access backhaul) in the deployment of 5G technology [17,40].

## 7. Conclusions

In this paper, the use of relay nodes in co-operative 5G networks has been studied. For this purpose, a decode-and-forward (D&F) relay node has been designed and implemented according to the latest 3GPP standard for co-operative 5G networks. The developed hardware platform is based on the use of software defined radio (SDR) and the Matlab${}^{\mathrm{TM}}$ tool, which presents great flexibility and reconfigurability for the implementation of potential 5G applications. Furthermore, the hardware developed for the implementation of the D&F relay node is deeply described by means of a flowchart following the algorithms implemented in Matlab${}^{\mathrm{TM}}$ software. A study of 5G NR downlink, in SISO and 2 × 2 MIMO configurations, has been carried out with signals complying with 3GPP standardized requirements. To evaluate the performance of the proposed architecture and its benefits, a set of measurements in a real outdoor-to-indoor environment have been carried out.

The obtained results clearly show that the D&F protocol co-operative network for 5G mobile wireless communications outperforms the non-co-operative architecture. The key performance indicators (KPIs), EVM, BER, and throughput obtained for the co-operative network are significantly better compared to those obtained for the direct link (gnodeB-to-UE). In this context, the proposed SISO D&F protocol co-operative network represents a substantial improvement, in terms of KPIs over 2 × 2 MIMO non-co-operative network. In addition, MIMO technology was able to reduce the bit error rate and increase the average capacity at the D&F relay node for 64-QAM and 256-QAM modulation schemes, demonstrating higher throughput and spectral efficiency of the co-operative network. For example, the maximum throughput of the proposed non-co-operative 5G network was 6.12 Mbps for the 256-QAM modulation scheme and MIMO antenna technique. Nevertheless, with D&F protocol, the obtained results from $P_{\mathrm{BS}}=9\;\mathrm{dBm}$ are higher than the direct link for the same configuration. On the other hand, we believe that the designed platform provides a cost-effective, scalable, and easy-to-upgrade solution to enable other 3GPP standardized 5G solutions and the application to wireless communications in millimeter bands, which is the future work to be addressed.

## Figures and Tables

**Figure 1 sensors-22-00913-f001:**
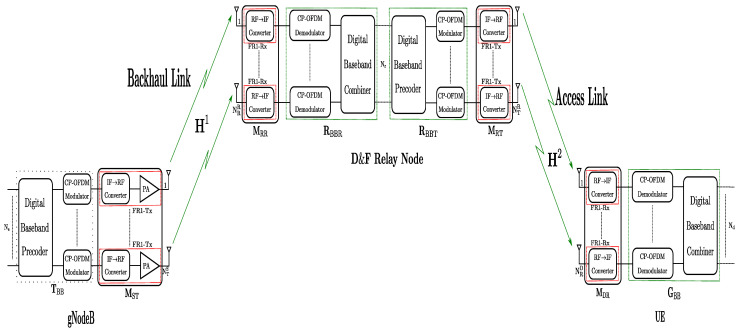
System model for 5G DL transmission via D&F co-operative architecture.

**Figure 2 sensors-22-00913-f002:**
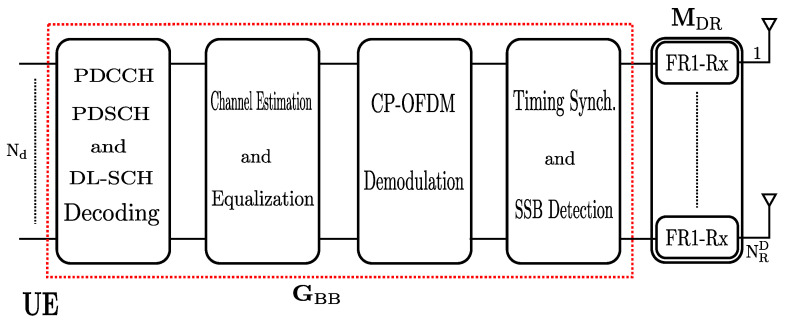
Baseband UE architecture.

**Figure 3 sensors-22-00913-f003:**
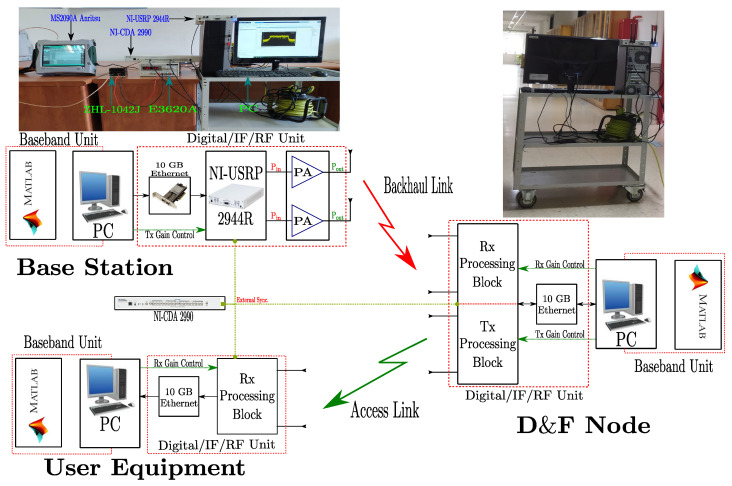
Hardware interface testbench for D&F relay node co-operative 5G network.

**Figure 4 sensors-22-00913-f004:**
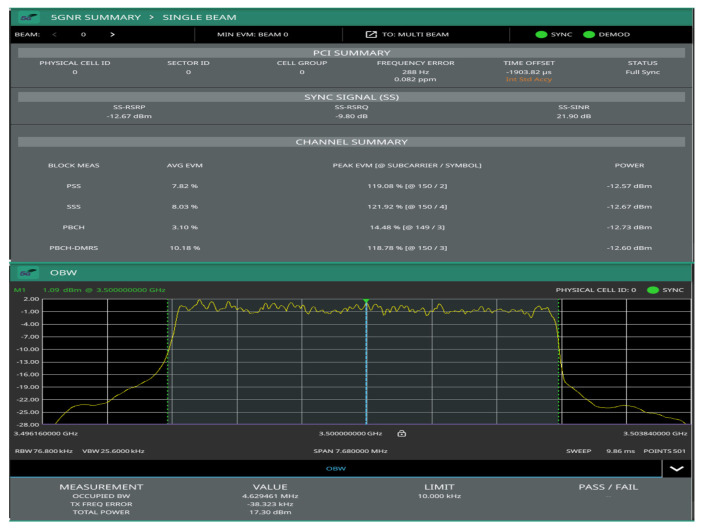
Transmitted signal demodulation, occupied bandwidth, channel power, and spectrum from the implemented gnodeB.

**Figure 5 sensors-22-00913-f005:**
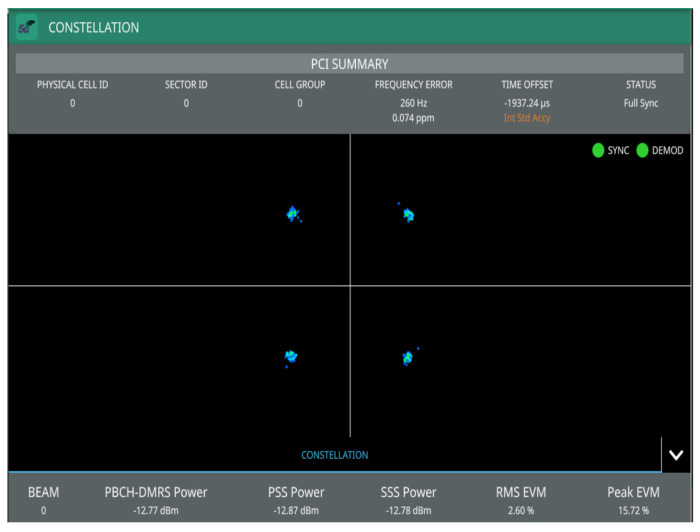
Q-PSK modulation scheme representation of the PBCH in the gnodeB measured through the MS2090A-Anritsu.

**Figure 6 sensors-22-00913-f006:**
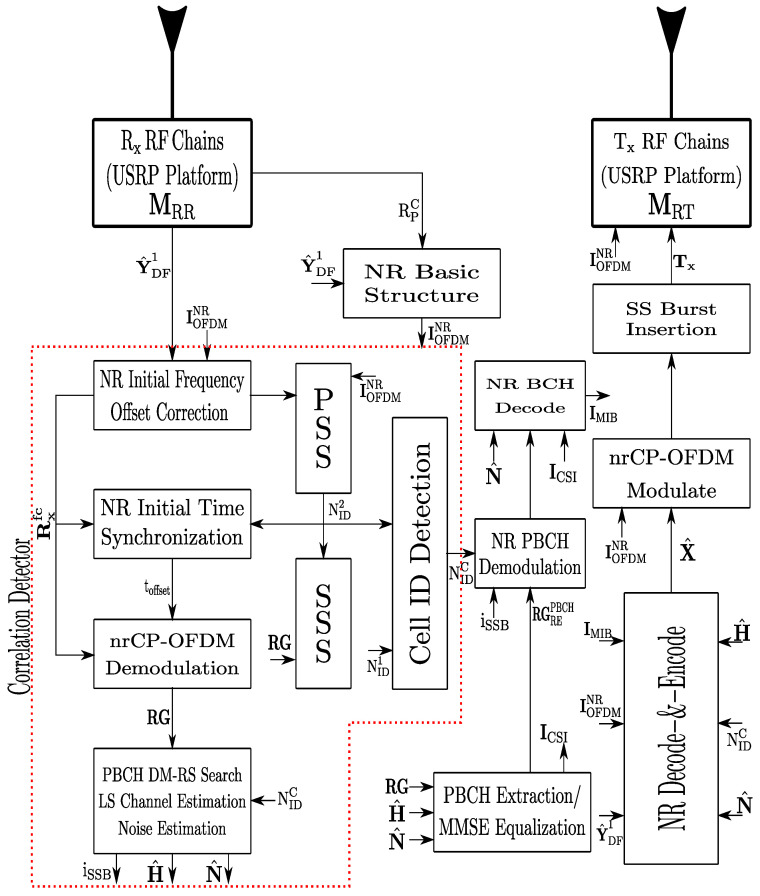
Proposed flowchart to hardware interface of 5G decode-and-forward relay node.

**Figure 7 sensors-22-00913-f007:**
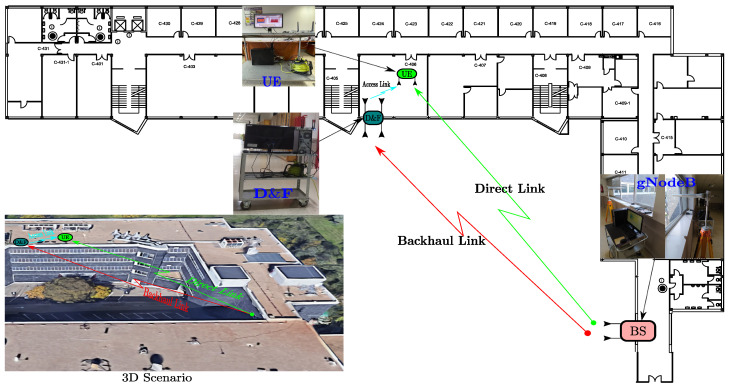
Overview of the testbench nodes and the experimentation scenario of the proposed D&F relay node co-operative 5G network.

**Figure 8 sensors-22-00913-f008:**
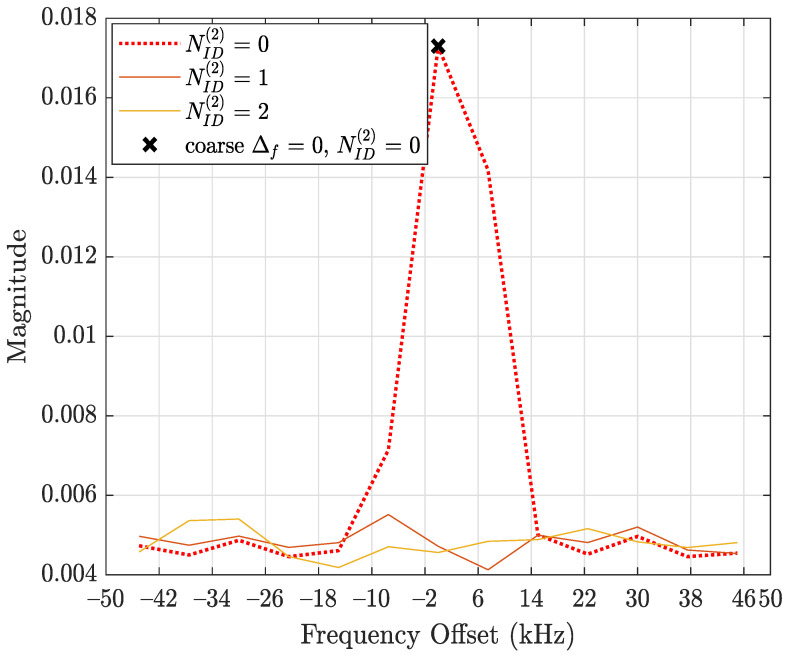
Primary synchronization signal correlations versus frequency offset.

**Figure 9 sensors-22-00913-f009:**
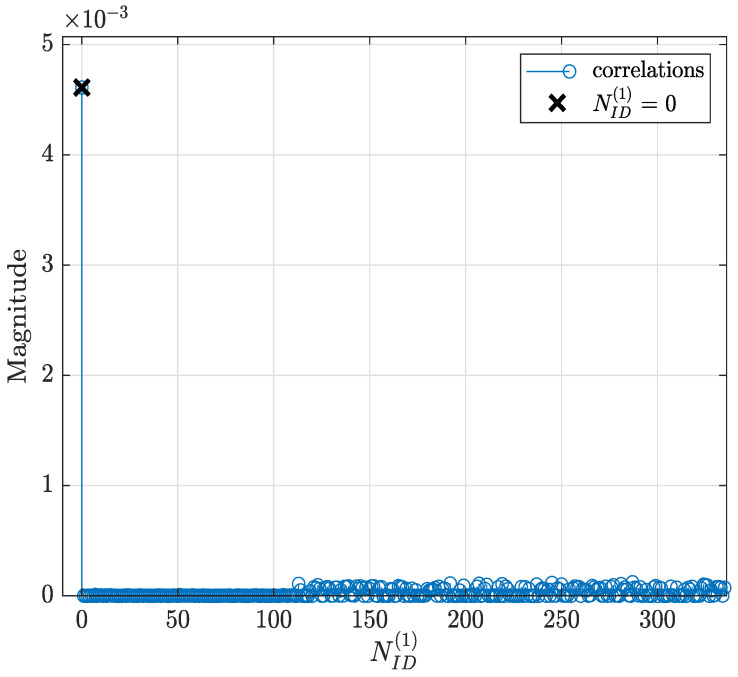
Secondary synchronization signal correlations.

**Figure 10 sensors-22-00913-f010:**
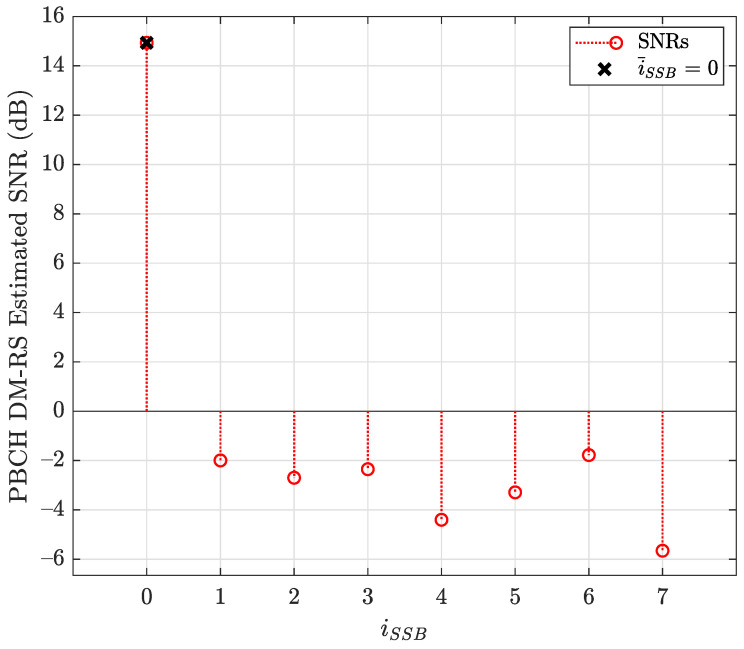
PBCH demodulation reference signal estimated SNR versus i${}_{\mathrm{SSB}}$.

**Figure 11 sensors-22-00913-f011:**
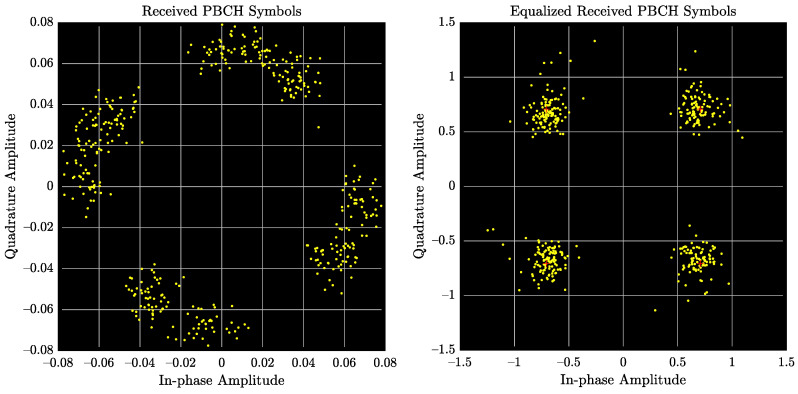
Received PBCH and equalized received PBCH constellation diagrams.

**Figure 12 sensors-22-00913-f012:**
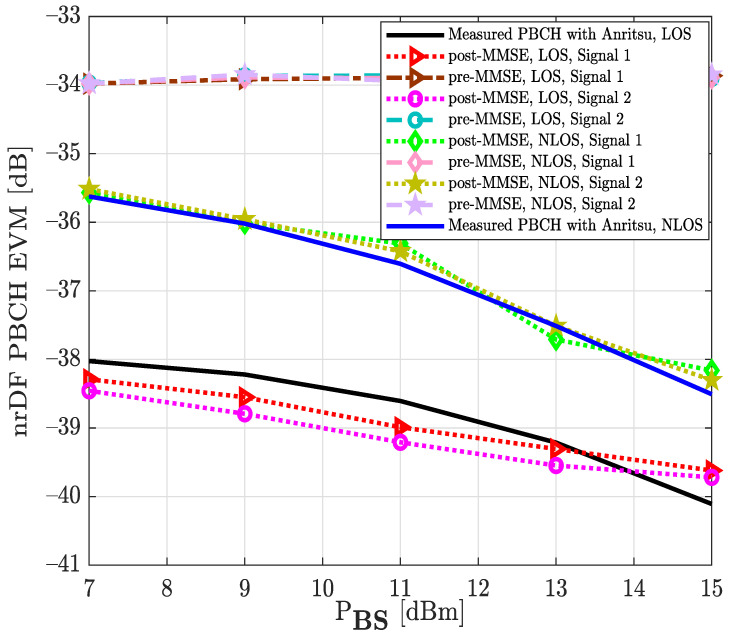
Median EVM at physical broadcast channel for two modulation schemes, LOS and NLOS environment, and equalization step under real test.

**Figure 13 sensors-22-00913-f013:**
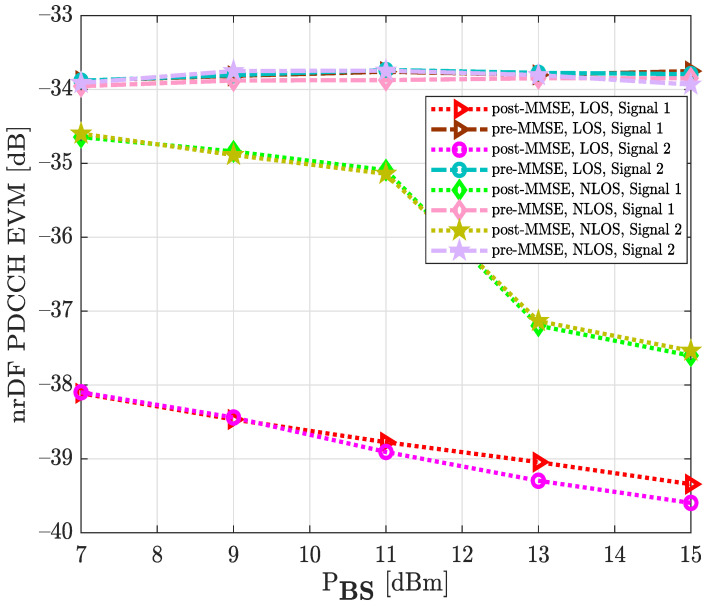
Median EVM at physical downlink control channel for two signals, LOS and NLOS environment, and equalization step under real test.

**Figure 14 sensors-22-00913-f014:**
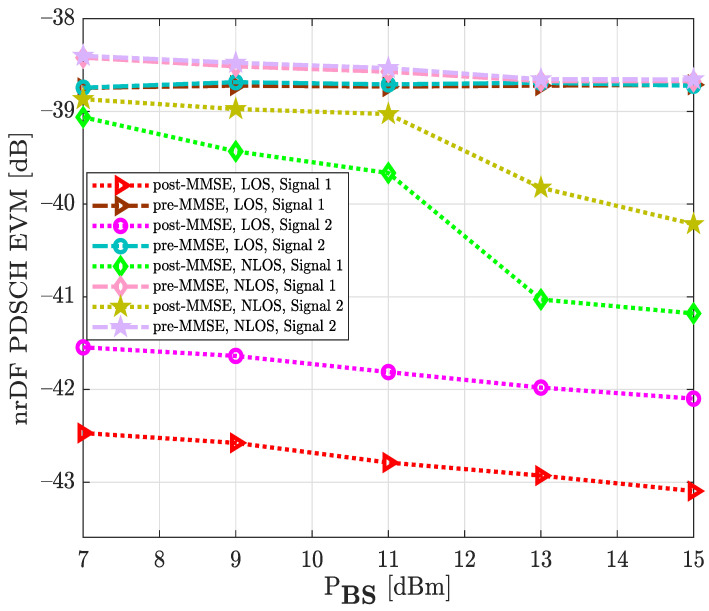
Median EVM at physical downlink shared channel for two modulation schemes, LOS and NLOS environment, and equalization step under real test.

**Figure 15 sensors-22-00913-f015:**
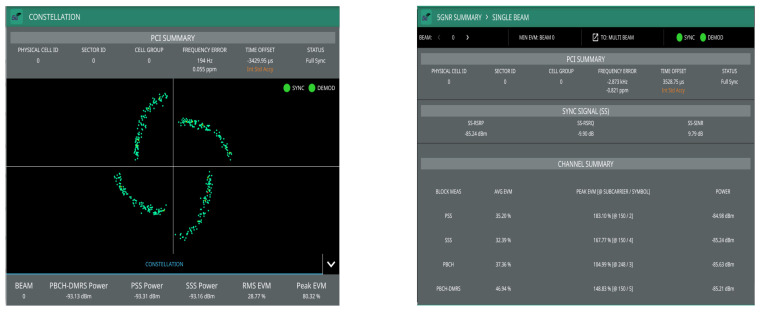
Demodulated and 5G summary of direct link though the Anritsu analyzer.

**Figure 16 sensors-22-00913-f016:**
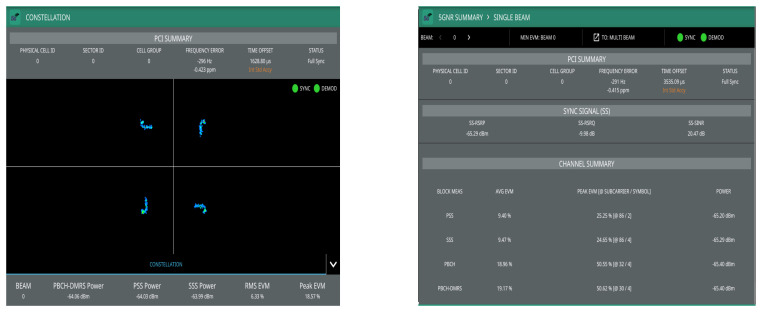
Demodulated PBCH and 5G summary of access link though the Anritsu analyzer.

**Figure 17 sensors-22-00913-f017:**
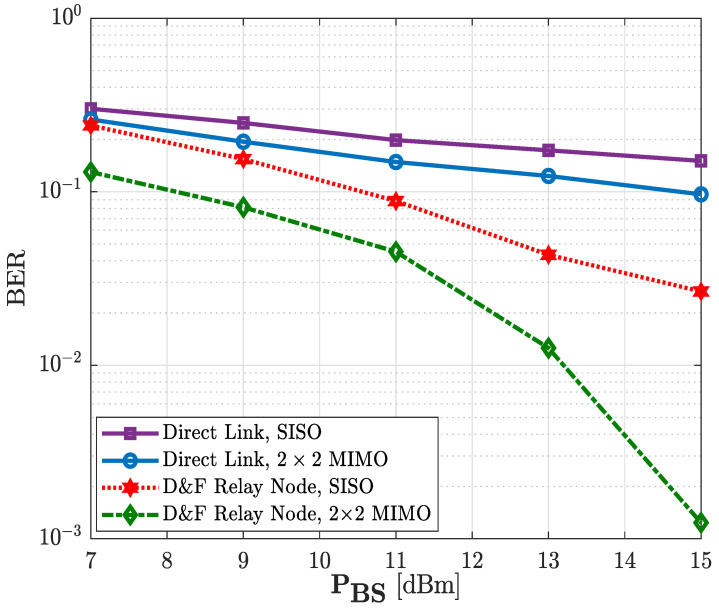
Bit error rate (BER) vs. transmission power of the gnodeB ($P_{BS}$) of the relay node co-operative 5G network; 64-QAM.

**Figure 18 sensors-22-00913-f018:**
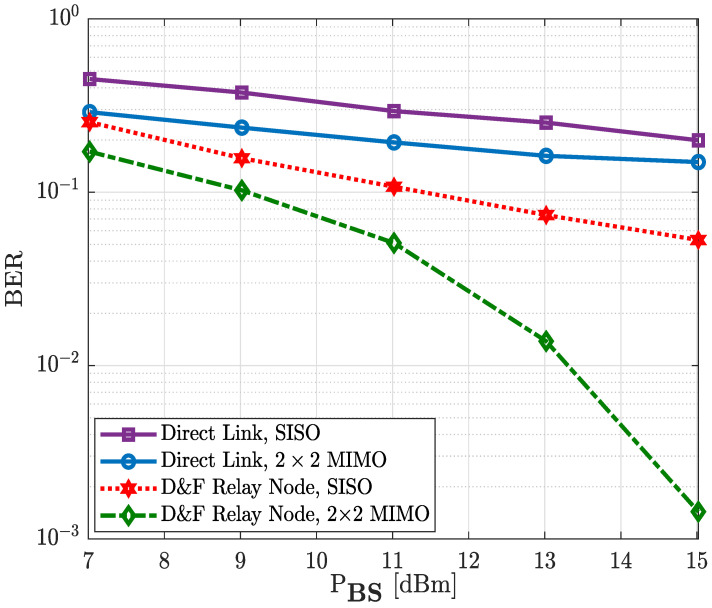
BER vs. $P_{BS}$ of the relay node co-operative 5G network; 256-QAM.

**Figure 19 sensors-22-00913-f019:**
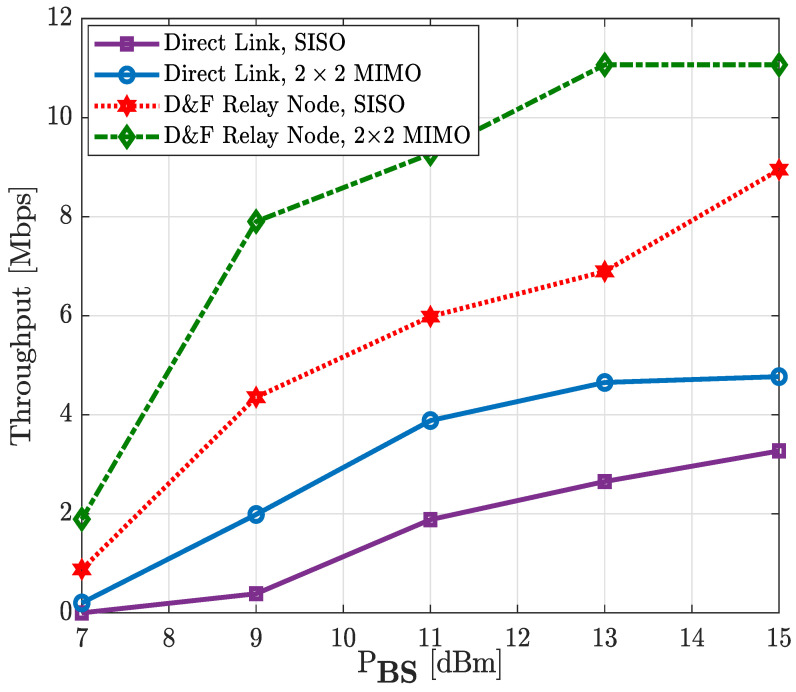
Throughput vs. $P_{BS}$ of the relay node co-operative 5G network; 64-QAM.

**Figure 20 sensors-22-00913-f020:**
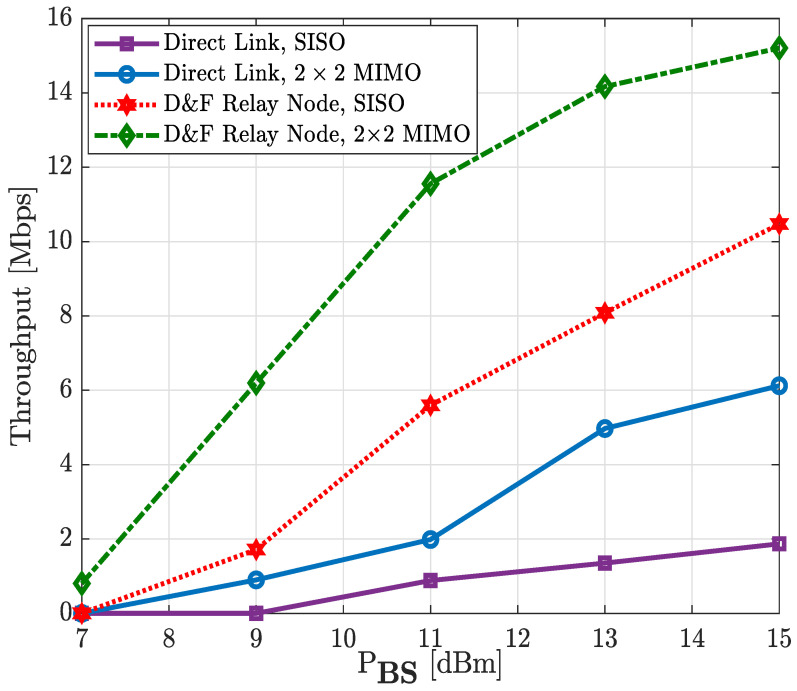
Throughput vs. $P_{BS}$ of the relay node co-operative 5G network; 256-QAM.

**Table 1 sensors-22-00913-t001:** Main parameters to emulate the gnodeB.

Parameters	Signal 1	Signal 2
Channel bandwidth, *F${}_{s}$*	5 MHz	5 MHz
Carrier frequency, *f${}_{s}$*	3.5 GHz	3.5 GHz
Tx/Rx schemes	SISO and 2 × 2 MIMO	SISO and 2 × 2 MIMO
5G duplex scheme	NR-FDD	NR-FDD
Modulation	64-QAM	256-QAM
Target code rate	3/4	4/5
Samples rate	7.68 MHz	7.68 MHz
Cyclic prefix (CP)	Normal	Normal
Number of FFT	512	512
Allocated resource blocks	25	25
Subcarriers per resource block	12	12
Information bit payload per slot	12,296	16,896
SS/PBCH block transmission	slot 0	slot 0
Subcarrier spacing	15 kHz	15 kHz
Number of cell ID	0	0
Max. throughput over 1 frame	11.066 Mbps	15.206 Mbps
Number of transmission	20	20
Number of frames per transmission	80	80

**Table 2 sensors-22-00913-t002:** Summary of results: With D&F relay node vs. direct link for $P_{\mathrm{BS}}=15\;\mathrm{dBm}$.

KPIs	Direct Link				D&F Protocol			
	SISO		2 × 2 MIMO		SISO		2 × 2 MIMO	
	64-QAM	256-QAM	64-QAM	256-QAM	64-QAM	256-QAM	64-QAM	256-QAM
BER	1.5 × 10${}^{-1}$	1.9 × 10${}^{-1}$	9.6 × 10${}^{-2}$	1.4 × 10${}^{-1}$	2.6 × 10${}^{-2}$	5.2 × 10${}^{-2}$	1.2 × 10${}^{-3}$	1.4 × 10${}^{-3}$
Throughput	3.27	1.87	4.77	6.12	8.95	10.47	11.06	15.20

## Data Availability

Data sharing not applicable.

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
