# Peer review of "Design and Synchronization Procedures of a D&F Co-Operative 5G Network Based on SDR Hardware Interface: Performance Analysis"

_sensors, 2022, doi:10.3390/s22030913_

Round 1

Reviewer 1 Report

Software Defined Radio dates back thirty years ago when Dr.Mitola coined the term

Software Defined Radio or Software Radio (SR) has become recently and more widely known as Cognitive Radios which is an even more intelligent version of SR

The Basic idea of an SR system is to digitize the wideband RF signal exactly next to the antenna (s)

In other words, in an SR all the functionalities for the bandpass signal should be software-based, so stills remains an extremely difficult task even if functionalities are realized in baseband as in the present work

This paper presents new experimental data and is very well structured. My comments can be summarized as follows:

-Hardware implementation is a little bit confusing since hardware (in terms of a new prototype hardware PCB platform is not implemented in this wok) so I suggest to the authors to use another term such as hardware configuration/connection/interface

-Figures should not exceed the width of the page

-Α discussion section before the conclusions with a summarization of the results as well as suggestions for future work is necessary

-EVM acronyms is not explained in the manuscript

-use "Several measurement cases have been carried out" instead of " An intensive measurement campaign has been carried"

-use "advantages" instead of "adventages"

Author Response

The authors wish to thank the reviewer for his comments and suggestions that will undoubtedly contribute to the improvement of the scientific quality of the article. 

Reviewer 2 Report

Reviewer: This paper proposes a Decode-&-Forward (D&F) Co-operative Hardware Network system which consists of an emulated base station processing unit (gNodeB), a D&F Protocol and the User Equipment (UE). The testbed is based on a Software Defined Radio (SDR) platform and Matlab software. They measure the EVM, BER and Throughput for signals with 64-QAM and 256-QAM modulation schemes. The obtained results show that the D&F Co-operative 5G Network achieves better performance in the communication between the gNodeB and the UE in an outdoor-to-indoor scenario. The reviewer has the following suggestions:

  • The authors should explain how does the estimated data symbols matrix come from.
  • Please check the marking of the x-coordinate in Figure 8.
  • Why does the EVM of the PBCH before the MMSE technique not decrease as transmitted power increases in Figure 11?
  • It is suggested to add more related work in the introduction, e.g., SWIPT Cooperative Spectrum Sharing for 6G-Enabled Cognitive IoT Network, Resource and Trajectory Optimization for Secure Communications in Dual Unmanned Aerial Vehicle Mobile Edge Computing Systems.

Author Response

The authors thank the reviewer for reading the manuscript and the comments made to improve the work. Special attention has been paid to the suggestions made and we hope that the revised version will meet your comments. Besides,
the authors have proofread the entire paper, and we have corrected typos and grammar issues.

Round 2

Reviewer 2 Report

The authors have well addressed the comments. It can be accepted.